# Multiplex flow magnetic tweezers reveal rare enzymatic events with single molecule precision

Rohit Agarwal [1,2] & Karl E. Duderstadt [1,2✉]

The application of forces and torques on the single molecule level has transformed our understanding of the dynamic properties of biomolecules, but rare intermediates have remained difficult to characterize due to limited throughput. Here, we describe a method that provides a 100-fold improvement in the throughput of force spectroscopy measurements with topological control, which enables routine imaging of 50,000 single molecules and a 100 million reaction cycles in parallel. This improvement enables detection of rare events in the life cycle of the cell. As a demonstration, we characterize the supercoiling dynamics and drug-induced DNA break intermediates of topoisomerases. To rapidly quantify distinct classes of dynamic behaviors and rare events, we developed a software platform with an automated feature classification pipeline. The method and software can be readily adapted for studies of a broad range of complex, multistep enzymatic pathways in which rare intermediates have escaped classification due to limited throughput.

[1] Structure and Dynamics of Molecular Machines, Max Planck Institute of Biochemistry, Martinsried, Germany. [2] Physik Department, Technische Universität München, Garching, Germany. ✉email: duderstadt@biochem.mpg.de

Force spectroscopy techniques have revolutionized functional studies of biomolecules, revealing key intermediates during protein folding[1], the mechanochemical cycles of polymerases and translocases[2,3], and the topological transformations catalyzed by DNA topoisomerases[4,5]. Most of these observations have been performed on instruments in which only one molecule can be studied at a time (i.e., optical tweezers), impeding the rapid collection of statistically significant datasets. Several approaches have been developed that provide greater multiplexing[6–9] including magnetic tweezers (MT)[4,10,11], acoustic force spectroscopy[7] (AFS), and centrifuge force microscopy[8] (CFM). However, all of these methods depend on high axial resolution to observe length changes. This imposes a fundamental limitation in throughput, requiring optics with a high numerical aperture (NA), which is only realized for small fields of view at high magnification. Novel approaches that overcome this throughput barrier are needed, so that rare events and unexpected behaviors can be distinguished from intrinsic heterogeneity in a statistically robust manner.

Here, we present a single-molecule DNA topology manipulation platform, flow magnetic tweezers (FMT), in which the experimental geometry used for magnetic tweezing is reconfigured by a lateral flow force. This transformation removes the high-NA limitation imposed by current force spectroscopy techniques allowing tether length changes to be observed as changes in lateral position in massive fields of view. To calibrate force solely based on lateral position measurements that report on changes in the projected length of individual DNA molecules, a system of equations, involving the equipartition theorem and worm-like chain model, was solved. This approach was independently validated using the known force and topology-dependent transitions of DNA.

To demonstrate the multiplexing capability of flow magnetic tweezers, we characterize the supercoiling dynamics and drug-induced DNA double-strand (dsDNA) break intermediates of gyrase. Gyrase is a type-II topoisomerase that plays an essential role in the maintenance of bacterial chromosomes by using the energy from ATP to conduct positive supercoil relaxation and negative supercoil introduction[12]. Using these activities, gyrase supports a wide range of essential cellular processes including transcription and DNA replication[13]. These vital functions of gyrase make it an ideal antibiotic target[12]. Several major classes of potent inhibitors target gyrase including quinolones like ciprofloxacin[14], which is commonly used to treat a wide range of bacterial infections. Structural and biochemical studies have demonstrated that ciprofloxacin kills gyrase function by stabilizing a DNA break intermediate, which prevents completion of the reaction cycle[12,15]. However, when and how this stalled intermediate is converted into an exposed DNA break that triggers a cellular response is not well understood. Using the improved throughput of flow magnetic tweezers, we establish links between break formation and topology-dependent enzymatic activity.

These experiments reveal a remarkable stability in gyrase-drug complexes and resistance to extreme torques.

To efficiently quantify the distinct types of gyrase behaviors and to cope with the vast quantities of data generated in each flow magnetic tweezer experiment, we developed a software platform called Molecule Archive Suite (Mars), which includes an automated feature classification pipeline. By combining this classification pipeline with a standard series of force and torque manipulations, we quantify a broad spectrum of gyrase behaviors. These include positive supercoil relaxation, negative supercoil introduction, unbraiding, and very rare DNA breaks. Studies of gyrase dynamics show the potential of FMT as a platform to rapidly characterize small subpopulations that exhibit rare enzymatic activities with single molecule precision. Multiplex flow magnetic tweezers can be readily adapted to studies of a broad range of complex, multistep enzymatic pathways in which throughput presents a significant challenge that prevents the discovery and characterization of novel intermediates.

## Results

**Flow magnetic tweezer construction and calibration**. To realize the multiplexing potential of combined flow and magnetic force transformations, a large 7× telecentric lens (TL), designed for industrial inspection of LCD monitors and microprocessors, was directly coupled to a 29 MegaPixel CCD camera generating a field of view greater than 15 mm² (Table 1 and Fig. 1a). Under typical experimental conditions this field contains around 50,000 beads (Fig. 1b) and offers the theoretical possibility of imaging more than 100,000 randomly arranged beads simultaneously (Supplementary Fig. 1). A combination of axial and lateral forces are applied to individual molecules tethered between a coverslip surface and a polystyrene bead at the base of a flow chamber. Vertical force ($F_m$) is applied by magnetic tweezers containing two large 1 cm block magnets in a horizontal configuration[16]. With optimal spacing, this arrangement creates a magnetic field parallel to the coverslip surface orienting the superparamagnetic beads, so that the rotation of the magnets can be used to apply positive and negative torque on the molecules. A lateral drag force ($F_d$) is applied to the beads using negative pressure at the outlet of the flow cell. A flow sensor provides continuous feedback and allows for programmable changes in flow with millisecond response time.

Multidimensional force control provides numerous avenues to characterize the physical properties of biomolecules in a massively parallel manner. For example, in the case of double-helical biopolymers like DNA, magnet rotation allows for the introduction of different topologies. Positive (counter-clockwise) rotation induces DNA overwinding and negative (clockwise) rotation induces DNA underwinding. In parallel, applied force can be modulated with flow, including complete reversal of flow direction, revealing the surface attachment location and projected length of individual molecules (Supplementary Movie 1). These transformations can be combined to study complex force-

---

**Table 1 Throughput of force spectroscopy techniques.**

| Method | Field of view (FOV) | Topological control | Resolution | Relative FOV | Tracked beads | Final |
|---|---|---|---|---|---|---|
| Flow magnetic tweezers (FMT) | 15 mm² (5.2 mm × 3 mm) | Yes | 21 nm | 3000× | 50,000 | 10,000 |
| Acoustic Force Spectroscopy (AFS)[7] | 0.89 mm² (1.1 mm × 0.8 mm) | No | 3 nm | 179× | 2000 | 150 |
| High-throughput magnetic tweezers (MT)[6] | 0.12 mm² (0.3 mm × 0.4 mm) | Yes | 3 nm | 24× | ~800 | Hundreds |
| Centrifuge Force Microscopy (CFM)[8] | 0.03 mm² (0.2 mm × 0.18 mm) | No | 2 nm | 7× | ~1000 | – |
| Magnetic Tweezers (MT)[41] | 0.005 mm² (83 µm × 64 µm) | Yes | – | 1× | 1 | 1 |

When the field of view was not provided, it was calculated using the magnification, pixel dimensions, and camera sensor size. Strick et al. used a 100× objective with a XC77CE Sony camera with 756 × 581 pixels and a pixel size of 11 µm × 11 µm which generates an estimated field of view of 83 µm × 64 µm. Yang et al. used a 40× objective with a Prosilica GC 2450 camera with 2448 × 2050 and a pixel size of 3.45 µm × 3.45 µm, which generates an estimated field of view is 211 µm × 176 µm. For this study, the number of mobile beads was determined using a flow reversal step. Resolutions were determined by tracking stuck beads and correcting for drift.

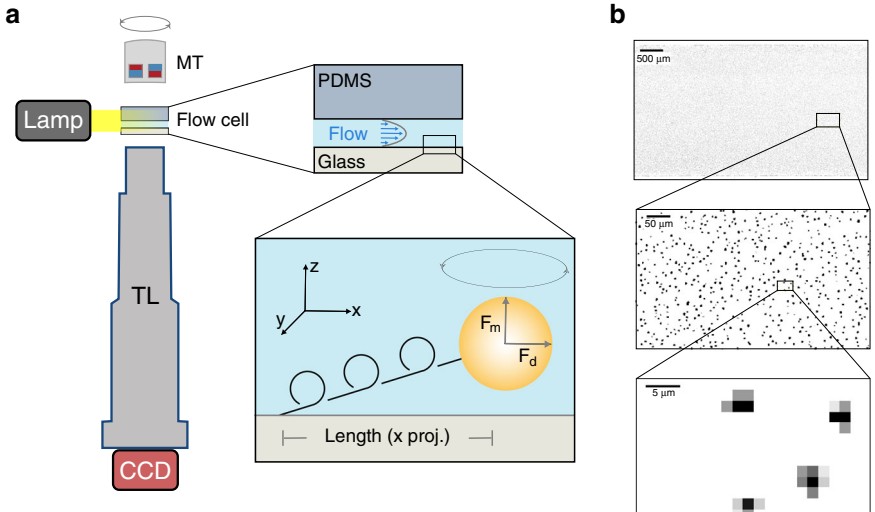

**Fig. 1 Principle of flow magnetic tweezers. a** The flow cell is mounted on a *x-y* translation stage with a fiber illuminator (Lamp). Beads are imaged with a 7× telecentric lens (TL) mounted on a 29 Megapixel CCD camera. Antiparallel magnets (MT) are mounted above the flowcell. The flowcell consists of a PEG-functionalized glass coverslip and a PDMS lid with an embedded flow lane. DNA molecules are tethered to the surface using biotin-streptavidin attachment. The total force on each molecule is the sum of the magnetic force (**F$_m$**) due to MT and drag force (**F$_d$**) due to flow. Magnetic beads tethered to the DNA molecules are locked in the magnetic field and linking number of the molecules can be manipulated by rotating the magnets. Manipulation of DNA length and topology can be monitored by tracking beads in the *x-y* plane. **b** A representative field of view (5.2 mm by 3 mm) from a single experiment containing 53,831 microspheres (diameter, 1 μm) enlarged by 10× and 100× to show individual microspheres.

extension behaviors. In Fig. 2a vertical force is held constant and magnet rotation is used to overwind and underwind DNA. Simultaneously, lateral force is increased in steps using flow. These transformations reveal an asymmetry in force-extension behavior as a function of topology[17]. At low force (<1 pN), both overwinding and underwinding of DNA leads to plectoneme formation and compaction in a symmetric fashion. Whereas, at higher forces (>1 pN), buckling is only observed in overwound DNA (Fig. 2b); for underwound DNA, local melting of the helix ensures minimal changes in length.

To characterize the force as a function of flow, we increased the flow velocity in steps and held the magnet position constant. The force can then be calculated with the equipartition theorem given by $F = k_B Tl/<\delta y^2>$, where $k_B$ is the Boltzmann constant, $T$ is the absolute temperature, $l$ is the length of the molecule, and $<\delta y^2>$ is mean-squared displacement of the beads in the direction orthogonal to flow (Fig. 1a). The other parameter required for force determination is DNA length. However, the improved throughput of FMT depends on tracking only lateral changes in position. As a consequence, only the projected length is observed. Therefore, both force and length were determined by assuming worm-like chain behavior and numerically solving using the mean-squared displacement $<\delta y^2>$ (Supplementary Fig. 2). Importantly, this numerical approach assumes observations were made using dsDNA and assumes a persistence length of 46 nm[18].

To independently validate our force calibration approach, we leveraged the topological control provided by the magnets and biophysical properties of DNA. The forces at which overwound and underwound DNA buckle are known[19]. Buckling of underwound DNA, shown in Fig. 2a, b, occurs at ~1 pN. Buckling of overwound DNA has a stronger dependence on the number of turns in the molecule. We evaluated the buckling transitions in molecules having 150 turns introduced corresponding to a sigma of 0.075. Under these conditions torque tweezers have estimated the positive buckling force to be ~6.5 pN[19]. To determine the flowrate for each transition, flow ramp experiments were performed in which flow was reduced or increased gradually (Fig. 3a). For both overwound

and underwound DNA, rapid buckling and extension was observed at distinct flowrates. Transition flowrates, 12.3 ± 0.2 μl/min (median ± s.e. median) for underwound DNA and 76.3 ± 0.8 μl/min (median ± s.e. median) for overwound DNA, were determined in an unbiased manner using a single kinetic change point detection algorithm (Fig. 3b, c)[20]. In Fig. 3d the flowrate versus force is shown as determined using numerical fitting together with the forces of the known buckling transitions.

To achieve very high throughput using flow magnetic tweezers some trade-offs were necessary. First, the dramatic increase in imaging area is only possible using a low magnification lens that results in a spatial resolution of 21 nm as determined by calculating the standard deviation of surface-immobilized beads after drift correction. This is 10-fold lower than other force spectroscopy methods (Table 1). Second, the large CCD camera needed to capture the very large field of view provides limited bandwidth. As a consequence, the highest frequency fluctuations of individual beads are not captured. A correction factor was applied to the bead fluctuation term to correct for over estimation of the force resulting from motion blur (Fig. 3d, see "Methods" section)[21]. The numerically determined correction factor ranged from 2% at lowest flowrate of 2.5 μl/min (used for all gyrase reactions) to 40% at the highest flowrate of 80 μl/min. Notably, the numerically determined force is in good agreement with the negative buckling transition (Fig. 3d, green point) in the low force region, but at the highest flowrates the force is lower than that predicted for the positive buckling transition (Fig. 3d, orange point). The force estimation at higher flowrates could be improved by using a higher bandwidth camera and increasing the magnetic force. These trade-offs also present challenges for studies involving short DNAs (less than 10 kb). The flow velocity approaches zero near the surface, so applying force on short DNAs would require very high flowrates. Moreover, projected length changes may not be observable due to the spatial resolution limit. DNAs that are 10 s of kbs in length are optimal for FMT experiments.

**Massive parallel imaging of topological transformations by gyrase**. Having established accurate force and topology control,

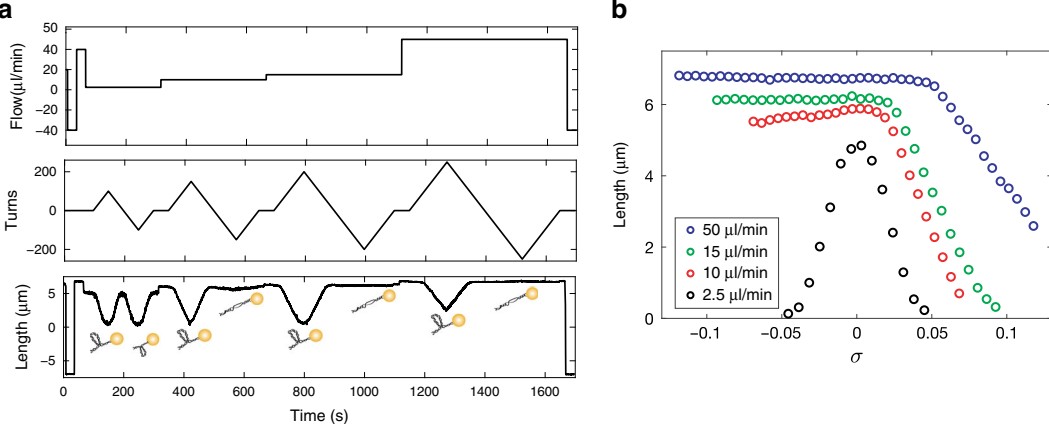

**Fig. 2 Force dependence of length for supercoiled DNA demonstrated by FMT. a** Changes in length are displayed for positive and negative supercoiling states at four flow rates. Negatively supercoiled DNA only compacts at the lowest flow rate, which corresponds to a force below 1 pN. The height of the magnets is held constant. **b** The mean length as a function of turns and force for a single molecule displays an asymmetric character, where positive buckling is less sensitive to increases in flowrate than negative buckling.

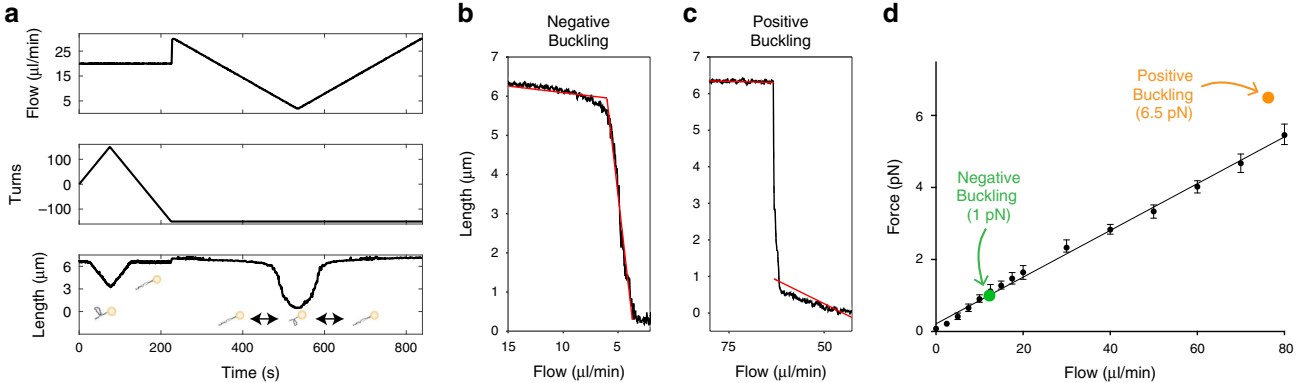

**Fig. 3 Force determination. a** A flow ramp experiment to determine the flow rate where negative buckling occurs. The molecule is singly-tethered and negatively supercoiled at high flow rate (middle panel). The flow is decreased gradually (upper panel) and a buckling transition is seen in the lower panel. **b**, **c** Two-segment kinetic change point fits to determine buckling flow rates for **b** negative buckling and **c** positive buckling transitions. **d** The force increases linearly as a function of flow (black points represent the median force). Error bars represent s.e. median. The median negative buckling transition flowrate from a single experiment is indicated in green (s.e. median is within the spot size) ($n = 399$ molecules). The median positive buckling transition from a single experiment is indicated in orange (s.e. median is within the spot size) ($n = 260$ molecules). The positive buckling transition is strongly dependent on torque. The measurement was conducted at 150 turns corresponding to a sigma of 0.075. The forces for each buckling transition were set to values from the literature for similar buffer conditions[19] (1 pN for negative buckling and 6.5 pN for positive buckling).

we next used FMT to investigate the kinetics of topological rearrangements by the topoisomerase gyrase. Prior to introduction of gyrase, individual tethers were classified using a series of changes in flow velocity and magnet rotations (Fig. 4a, b and Supplementary Movie 2). First, flow was reversed to identify mobile beads and the attachment site of each tether. Second, negative and positive coiling steps were conducted at high force to identify coilable molecules and distinguish them from multiply tethered beads. Third, after force determination at constant flow and magnet position, negative and positive coiling was performed a second time to determine the rate of compaction versus turns at the reaction flow rate. Finally, the tethers were left positively coiled ($\sigma > 0$) and gyrase was introduced. Upon arrival of gyrase, tethers rapidly extended and, following a pause, compacted at a slower rate. As expected for topoisomerase activity, the behavior was not observed in the absence of ATP or on nicked, noncoilable molecules (Supplementary Fig. 3). Our observations are consistent with previous reports showing that gyrase resolves positive supercoils and has the unique ability to introduce negative

supercoils on relaxed DNA molecules[4]. Therefore, we interpret the extension as resolution of positive supercoils and the compaction as introduction of negative supercoils. Since each enzymatic cycle of gyrase is known to change the net linking number (Lk) of DNA by −2 turns[22], we conclude that each burst of activity represents multiple enzymatic cycles.

To further investigate the force and topology dependence of gyrase activity, we took advantage of the information-rich manipulations performed with FMT prior to gyrase arrival. Molecule-by-molecule force determination revealed differences in the applied force on individual molecules within single experiments. We attribute this distribution of forces to differences in the size and magnetic content of individual microspheres. Additionally, depending on buffer conditions and sample preparation, bead clusters were observed, resulting in increases in force proportional to cluster size. Using current force spectroscopy techniques these subpopulations may not be statistically significant, but the massive multiplexing of FMT allows for their quantification in studies of complex activity versus force relationships within single experiments.

Using this approach, we determined the force dependence of gyrase activity in just two experiments performed with different flow velocities (Fig. 4c). We observed a peak velocity for positive relaxation of 1.26 ± 0.07 cycles/s (median ± s.e. median) at 0.2 pN after which the velocity decreases to a plateau of ~0.85 cycles/s. Introduction of negative supercoils is more sensitive to force having a peak velocity at 0.65 ± 0.05 cycles/s (median ± s.e. median) at ~0.2 pN and loss of activity above ~0.5 pN. The force–velocity relationship we observe qualitatively agrees with observations made using magnetic tweezers[4]. Our peak velocity also quantitatively agrees with past reports[4,5]. We interpret the force and topology dependence of gyrase we observe as a combination of distinct operational modes previously reported. Below ~0.5 pN is α mode, where gyrase resolves positive supercoils by strand passage of a proximal segment and introduces negative supercoils. Above ~0.5 pN gyrase is in χ mode for which only positive supercoils are resolved by strand-passage of a distal segment[4,5].

To efficiently process the large datasets generated in FMT experiments we developed Mars—a high-performance collection of single-molecule analysis plugins for Fiji[23–25] written in Java (source code and installation instructions available at https://github.com/duderstadt-lab). Using Mars, we were able to locate all beads, track them with subpixel accuracy, and develop secondary classification algorithms based in part on kinetic change point analysis[20] (Fig. 4a). The tracking results from FMT experiments typically do not fit in physical memory. Therefore, Mars relies on a new core data model based on collections of molecule objects each with a universally unique identifier (UUID) that are retrieved from the hard drive on demand. The architecture allows for seamless virtual storage, merging, and multithreaded processing of very large datasets.

We benchmarked the multiplexing capability of flow magnetic tweezers by analyzing a gyrase experiment in which 53,831 molecules were tracked. Among those, the feature classification pipeline of Mars identified 12,832 coilable molecules and a final set of 8874 accepted molecules, which were also reversible, singly tethered and remained so throughout the experiment. Mars reported 7969 bursts of positive relaxation and 6183 bursts of negative introduction. The massive throughput achieved is visible in the velocity versus burst time scatter plot in Fig. 4d. FMT provides more than an order of magnitude higher throughput than existing force spectroscopy approaches (Table 1).

**Gyrase inhibition and DNA break formation induced by ciprofloxacin.** The improvement in throughput offered by FMT allows for rapid characterization of numerous subpopulations from individual experiments and direct observations of rare, yet critically important, events in the life cycle of the cell. To demonstrate the detection power of FMT, we performed a series of experiments to directly visualize gyrase dynamics in the presence of the potent inhibitor ciprofloxacin. During each cycle of activity, gyrase makes a transient double-strand break in one DNA strand, passes a second strand through, and then reseals the break[26]. The transient double-strand break formed by gyrase during strand passage is exploited by ciprofloxacin, which prevents resealing of the break upon binding (Fig. 5a). The drug is believed to kill cells by at least two pathways. In one pathway, inhibition of relaxation of positive supercoils and introduction of negative supercoils is believed to shutdown DNA replication and transcription[12]. In the other pathway, the transient DNA breaks stabilized by ciprofloxacin are thought to be converted into permanent DNA breaks that cause severe genome damage[27]. The mechanism of the former pathway is clear, but how and when gyrase-drug complexes disassemble to become *bona fide* DNA

breaks and reveal themselves to the cell is not well understood. As a consequence, the sequence of events during the cellular response to many important antibiotics remains unclear.

Gyrase reactions were conducted in the presence of increasing concentrations of ciprofloxacin to characterize inhibition and detect DNA breaks. Importantly, the sequence of transformations outlined in Fig. 4b was conducted for all experiments so molecules could be filtered using the feature classification pipeline outlined in Fig. 4a. This filtering step ensures the gyrase dynamics reported are only from properly formed, non-nicked DNA tethers. As previously outlined, gyrase was introduced to positively supercoiled DNA molecules and the rates from bursts of positive supercoil relaxation and negative supercoil introduction were determined using sliding windows. The burst analysis revealed a dose-dependent reduction in gyrase activity (Fig. 5b, c and Supplementary Fig. 4) with an IC50 of 0.94 μM (0.2–4.1; 95% CI) for positive supercoil relaxation and 5.3 μM (2.6–12.1; 95% CI) for negative supercoil introduction. These values are consistent with past estimates placing the IC50 around 0.5 μM using gel-based cleavage assays[28].

Given that ciprofloxacin stabilizes a transient DNA break within gyrase, we expected to see an increase in DNA breaks as a function of drug concentration. However, surprisingly, the frequency of DNA breaks did not increase even at drug concentrations of 20 μM and above with severely reduced activity. Nonetheless, rigorous classification of filtered molecules did reveal very rare DNA break events (Fig. 4a and Supplementary Fig. 5). In total 26,063 molecules passed all filters in ten experiments conducted at a range of ciprofloxacin concentrations and 26 DNA breaks were observed in the presence of gyrase. The breaks were evenly distributed among the ciprofloxacin concentrations. We cannot entirely exclude the possibility that these breaks arise due to nuclease contamination in the gyrase protein stock. However, DNA breaks were three times more frequent during activity bursts than at random locations during the period of gyrase incubation (Fig. 5d). Frequencies were compared by considering the number of burst windows during the observation period. DNA breaks occurred in only 1 in 1000 molecules and were only detectable due to the massive throughput improvement of FMT.

**Visualization of 100 million reaction cycles in parallel.** To further probe the enzymatic activity of gyrase and surprising stability of gyrase-drug complexes, a treadmilling assay was developed to visualize large numbers of reaction cycles in individual experiments. In the treadmilling assay (Fig. 5e), the magnet is rotated continuously after completion of the standard transformation series shown in Fig. 4b. Three consecutive 30 min periods, with increasing magnet rotation rates, revealed a robust ability of gyrase to continuously resolve positive supercoils as they were introduced (Fig. 5f, g and Supplementary Fig. 6). At the lowest rate of two rotation per second or one cycle per second, gyrase not only resolved the positive supercoils but introduced negative supercoils in bursts (blue trace, Fig. 5g). At higher rotation rates gyrase was only able to remove positive supercoils as they were introduced. Formation of positive supercoils at the highest rate of eight rotations per second or four cycles per second, triggered the recruitment of additional gyrase molecules similar to the accumulation of gyrase visualized ahead of DNA replication[29]. On average 7801 molecules passed filtering in each of three treadmilling experiments performed and 98 ± 11 million gyrase reaction cycles were visualized at the single-molecule level during each 3-h experiment. Using FMT, single-molecule imaging of billions of reaction cycles is easily within

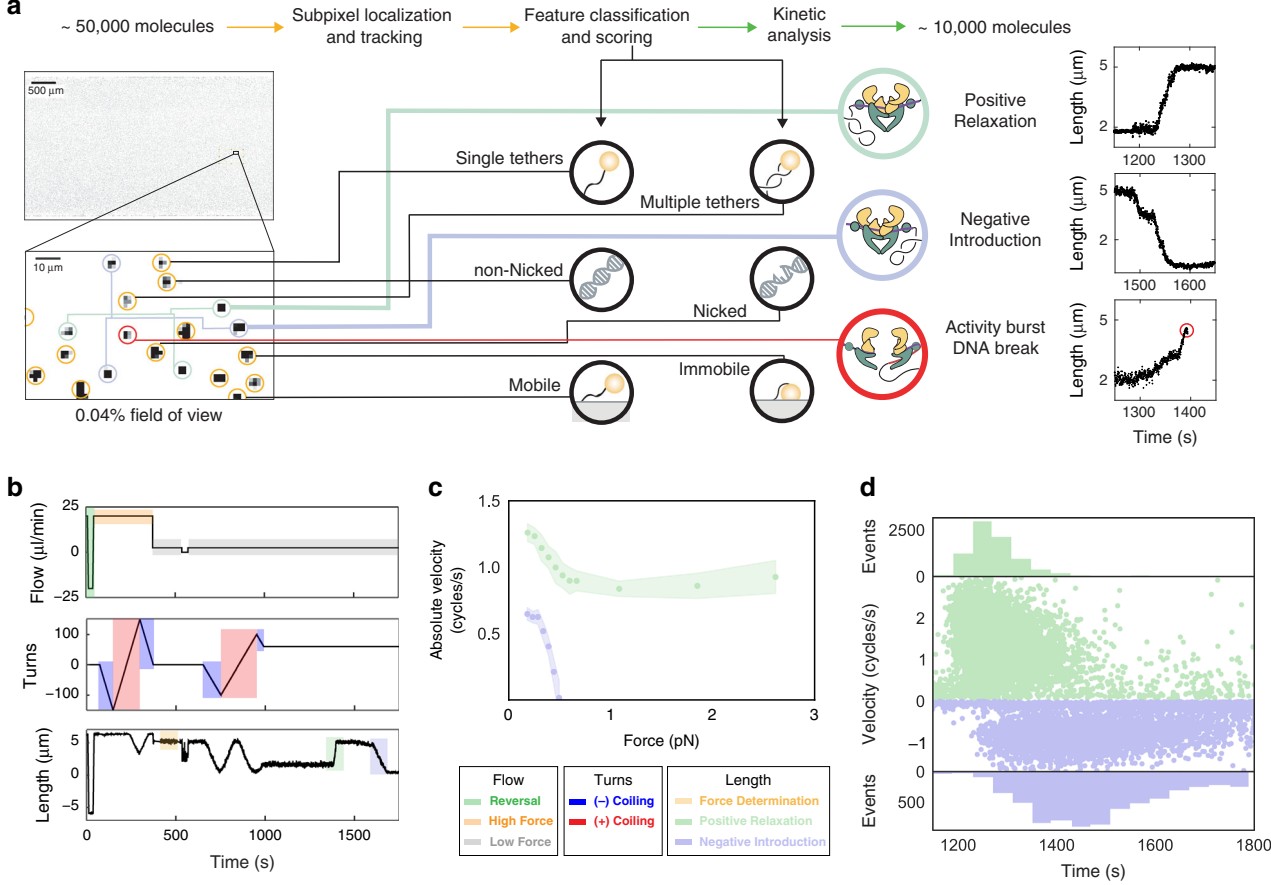

**Fig. 4 Massive parallel imaging of DNA supercoiling by gyrase. a** Automated feature classification pipeline to convert raw videos to annotated Molecule Archives containing scores for standard features and tags for gyrase behaviors. Images and example traces were selected from three independent experiments. **b** Changes in flow and magnet rotations used for molecule classification. Mobility is checked using flow reversal (green). Negative and positive coiling are conducted to select singly-tethered beads (blue and red). Force is determined at higher frame rate (33 Hz, orange). Finally, rapid lengthening, pausing, and shortening are observed as gyrase relaxes positive supercoils (green) and introduces negative supercoils (blue). **c** Velocity (cycles/s) versus force (pN) dependence of gyrase. Median rates of positive supercoil relaxation (green) and negative supercoil introduction (blue) ($n = 680$ molecules from two independent experiments). Shaded regions represent the s.e. median. **d** Gyrase activity burst velocities (cycles/s) versus start times (s) for individual molecules observed in a single experiment. Bursts of positive relaxation (green, $n = 8426$ bursts from one experiment) and negative introduction (blue, $n = 3302$ bursts from one experiment). Molecules at higher forces did not have negative supercoil introduction bursts, but were nonetheless plotted and show up as a broad scatter near zero. Of the 53,831 microspheres tracked, 12,172 passed the pipeline outlined and were evaluated for activity bursts.

reach if experiments are conducted for longer than 3 h with high rotation rates.

The treadmilling assay provided the opportunity to probe the stability of gyrase-drug complexes in response to torque. Treadmilling experiments were conducted in the presence of increasing concentrations of ciprofloxacin to characterize inhibition of treadmilling activity and detect whether DNA breaks form as a result of positive or negative torque. At high drug concentration gyrase was inhibited and DNA rapidly compacted in response to magnet rotations (Fig. 5g). Remarkably, no increase in DNA break frequency was observed as a function of ciprofloxacin for high levels of positive or negative torque (up to $\sigma = 0.35$). To confirm that gyrase-drug complexes housing transient DNA breaks were in fact formed under our experimental conditions, an extreme 0.1% SDS wash step was introduced after completion of the torque step. The detergent wash revealed massive, dose-dependent DNA breaks as gyrase-drug complexes unfolded confirming their formation in the torque experiments (Fig. 5h). Importantly, a control in the absence of gyrase and drug revealed no tether loss during SDS washing. These observations demonstrate the extreme stability of

gyrase-drug complexes over a wide range of experimental conditions.

## Discussion

Flow magnetic tweezers is a single-molecule technique for studying complex force versus topology relationships in biomolecules and biomolecular systems in a massively parallel manner. The multiplexing capability of FMT allows for statistically robust quantification of small subpopulations, rare events, and complete force characterization in single experiments. The detection power of FMT was demonstrated by investigating the supercoiling dynamics of gyrase in the presence and absence of ciprofloxacin. To rapidly analyze the massive amounts of data generated during each FMT experiment, we developed a new software platform (Mars) with an automated feature classification pipeline. In the case of gyrase, analysis using this software revealed a broad spectrum of activities including positive supercoil relaxation, negative supercoil introduction, unbraiding, and rare DNA breaks.

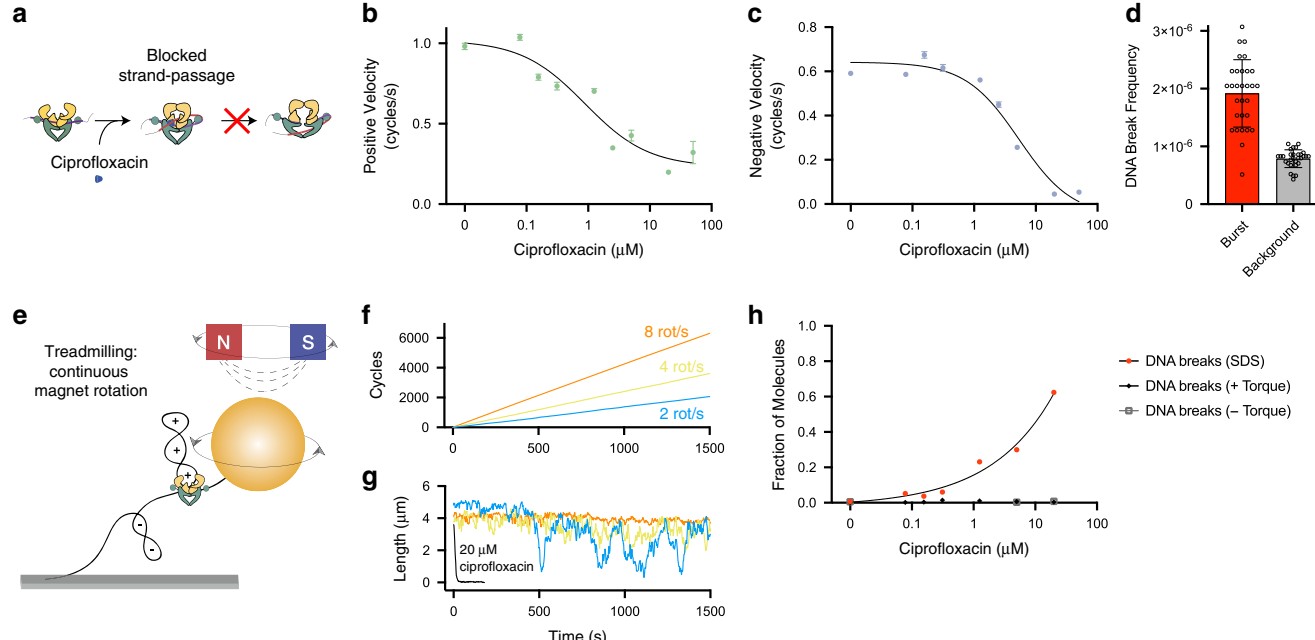

**Fig. 5 Gyrase inhibition and DNA break formation due to ciprofloxacin. a** Enzymatic cycle of gyrase. Gyrase binds a gate segment, makes a transient DNA break, and passes a transfer segment through. Ciprofloxacin prevents resealing of the break. **b** The velocity of positive supercoil relaxation as a function of ciprofloxacin concentration. The IC50 from the inhibitor versus response equation ($Y = Bottom + (Top\text{-}Bottom)/(1 + (X/IC50))$) was 0.94 μM (0.2–4.1; 95% CI). **c** The velocity of negative supercoil relaxation as a function of ciprofloxacin concentration. The IC50 from the inhibitor versus response equation ($Y = Bottom + (Top\text{-}Bottom)/(1+(X/IC50))$) was 5.3 μM (2.6–12.1; 95% CI). Error bars represent the s.e. median. The total burst counts for individual experiments are provided in the "Statistics and reproducibility" section. **d** Frequency of DNA breaks that occur during bursts of gyrase activity (red) compared to background (gray) normalized by the mean burst length ($n = 26{,}063$ molecules from ten independent experiments). In total 26 breaks were observed with eight occurring during bursts (Supplementary Fig. 5). Error bars represent s.d. determined by 30 cycles of bootstrapping. **e** Treadmilling assay developed to observe continuous gyrase activity for long periods. The magnet is continuously rotated to introduce positive supercoils. Gyrase continuously resolves the positive supercoils as they form (Supplementary Fig. 6). **f** Number of gyrase reaction cycles performed as a function of time from a single molecule for different rotation speeds. **g** DNA length changes during the treadmilling assay. Loss of activity and rapid compaction in the presence of ciprofloxacin is shown in black for eight rotations per second. **h** Fraction of molecules with DNA breaks as a function of ciprofloxacin concentration, positive and negative torque, and SDS treatment.

Gyrase-drug complexes are remarkably stable. No dose-dependent increase of DNA breaks was observed under normal buffer conditions or in response to extreme positive or negative torque for 10 s of thousands of molecules. The improved detection power of FMT did reveal rare DNA breaks during bursts of gyrase activity, however, these occurred with equal frequency independent of drug concentration. The transient DNA breaks stabilized by ciprofloxacin were only converted into irreversible breaks that triggered bead loss in the presence of detergent. The stability of gyrase-drug complexes under extreme torques suggests that the positive supercoils generated ahead of an advancing transcription or replication complex are not sufficient to generate breaks. Instead a physical encounter is likely required to convert transient breaks into irreversible breaks[30] that are ultimately exposed to the cell during failed repair attempts[31]. The improved throughput of FMT allows for direct characterization of the dynamics of encounters of complex machineries[32], which were previously out of reach.

Characterization of other major classes of antibiotics and anticancer drugs[12] that target both the bacterial and eukaryotic topoisomerases promise to clarify the biophysical properties of critical intermediates in the pathways of drug inhibition and DNA break formation. FMT and the software platform developed can be easily tailored to studies of a broad range of enzymes that conduct nucleic acid transactions. Given that FMT can be implemented using low cost components on standard microscopes, we anticipate applications in a broad range of fields. Moreover, implementing FMT on a traditional magnetic tweezer setup requires only small modifications but results in a substantial improvement in throughput.

## Methods

**Experimental setup**. All single-molecule force and topology studies were conducted on a setup constructed as follows. A 29-megapixel charge-coupled device (29 M Prosilica GX6600 GigE Monochrome camera, Allied Vision 6576 × 4384 pixel dimensions) was mounted directly to a 7× telecentric line scan lens (TL12K-70-15, Lensation GmbH, numerical aperture of 0.23)[33] generating a field of view of ~15 mm². Each pixel represented 0.78 μm × 0.78 μm at 7× magnification and 1× binning. The Lens-Camera assembly was mounted on a vertical translation stage (Newport, M-MVN50) for focusing and positioned below an XY-translation stage (Physik Instrumente M-545.2ML) for sample alignment. A custom-machined holder was used for mounting on the stage and sealing a PDMS flow cell. The flow cell consisted of two inlets and one outlet tube (Supplementary Fig. 7), where the latter was attached to an appropriate flow sensor: BFS, Microfluidic Coriolis flow sensor, Bronkhorst-Elveflow or MFS3, Microfluidic Thermal flow sensor, both purchased from Elveflow and connected to an OB1 pressure control system. The two inlets allow for fast switching between buffers without disrupting buffers. Negative and positive pressure was applied using vacuum and nitrogen outlets. Flow profiles were programmed and controlled using ESI microfluidic software from Elveflow. The magnetic tweezers were mounted above the flow cell and consisted of two antiparallel 1 cm neodymium (supermagnete, W-10-N) block magnets placed 6 mm apart. Vertical translation and rotation of the magnets was achieved using dc servo motors (M-126.PD1, C-150.PD, 2× C863, Physik Instrumente) mounted on a rail tower for course vertical alignment (X95, M-CSL95-80, Newport) with an additional linear stage for centering the magnets over the sample (PT1/M, Thorlabs). Additional posts, and clamps for mounting, were either custom machined in house, or purchased from Thorlabs or Newport. Magnets were kept at a constant height above the flow cell for all experiments and only changes in flow were used to alter applied force on the beads. A fiber illuminator (OSL2, OSL2FB, OSL2COL, Thorlabs) was setup laterally to the flow cell to illuminate the beads in a

dark field. Videos were acquired using StreamPix 6 software suite (Norpix) typically at 4 or 33 Hz with 2× binning enabled to improve readout performance.

**Flow cell setup**. Glass coverslips (60 mm × 24 mm) were functionalized as previously described[33]. They were placed inside a plasma cleaner (Zepto B, Diener electronic) and cleaned under occasional influx of oxygen for 20 min. The surface was silanized by incubating the plasma-cleaned slides in a 2% v/v solution of freshly opened APTES (aminopropyltriethoxysilane, ROTH) in acetone. A solution of inert and biotinylated PEG (MPEG-SC-5000-5g, Biotin-PEG-SC-5000-1g, Laysan Bio Inc.) in the ratio 12%:1% (w/v) was dissolved in 0.1 M NaHCO₃ at pH of 8.2. The PEG solution was put between two stacked silanized slides overnight, after which the slides were washed, dried, and stored in vacuum for up to a month at a time. Prior to experiments, slides were incubated with 0.2 mg/ml streptavidin (Streptavidin from Streptomyces Avidini, Sigma–Aldrich) in 1× working buffer[33]. A PDMS cover with an embedded lane was placed on the coverslip and sealed using a frame built into the flow cell holder (Supplementary Fig. 7). The inlet tubes had an inner diameter of 0.58 mm and an outer diameter of 0.96 mm (PE tubing PORTEX, SX05, laborversand.de) and the outlet tube was 1/32" ID PTFE tubing (Darwin microfluidics). Following the setup of the flow cell on the microscope, the cell is flushed with the given buffer (1× PBS for force calibration, buckling and supercoiling experiments and 1× Gyrase buffer for the enzymatic experiments) and incubated for at least 30 min. The topologically constrained DNA (5 pM), with biotins on one end and digoxigenins on the other (DNA preparation), was introduced to the flow cell at a constant flow rate for 30 min to allow time for attachment to streptavidin molecules on the surface. After incubation with the DNA, antiDIG fab fragment-coated beads were flushed into the flow cell for attachment to the exposed digoxigenin end. Once sufficient bead attachment is observed, excess beads are washed out of the flow cell and the magnet is lowered into position. Mobile tethers are then identified using a flow reversal step (Supplementary Movie 1) and coilable tethers were identified with a series of coiling steps using the magnets (Supplementary Movie 2). The buffer used for supercoiling, force calibration, and buckling experiments contained 0.01 M phosphate buffer, 2.7 mM KCl, 0.137 mM NaCl (1× PBS tablet solution from Sigma), 0.1 mg/ml BSA, and 0.1 % TWEEN20. Gyrase reaction buffer contained 35 mM Tris-HCl (pH 7.5), 24 mM potassium glutamate, 4 mM MgCl₂, 2 mM DTT, 0.2 mM spermidine, 1 mM ATP (ATPγS), 6.5% (v/v) glycerol, 0.1 mg/ml BSA, 0.1% TWEEN20, and indicated amounts of DNA gyrase, typically 0.2 nM[4]. Where indicated ciprofloxacin (17850-5G-F, Sigma) was added at the concentrations specified.

**DNA and bead preparation**. The torsionally constrained DNA constructs were ~21 kb in length, which corresponded to a contour length of 6.8 µm. They were generated using the plasmid Supercos1-Lambda 1,2 generated in *E. coli* cells as previously described[34]. To avoid damaging the final DNA substrate (20,666 bps), a Chargeswitch Plasmid miniprep kit (Invitrogen) was used to extract the plasmid. The plasmid was then double-digested using NotI and XhoI sites and purified again using a Chargeswitch PCR clean-up kit (Invitrogen). Once digested, the segments were ligated (T4 DNA ligase, NEB) to DNA linker fragments (~550 bp) that had biotinylated and digoxigenated dUTP moieties incorporated via PCR[35]. The beads (1 µm Dynal MyOne, Invitrogen 650-01) were prepared as follows. They were washed in 0.1 M Borate buffer (H₃BO₃) at pH 9.5 and incubated under constant overnight shaking with DIG antibody Fab fragments (Roche diagnostics GmbH) in PBS at 37 °C. Excess fragments were washed away using Tris-HCl buffer and stored for months in the same at 4 °C[33]. The bead stock was sonicated for 1 min and mixed prior to dilution in the experimental buffer (either PBS or gyrase buffer).

**Microsphere tracking and analysis software**. To efficiently process the large datasets generated in FMT experiments we developed Mars—a high-performance collection of single-molecule analysis plugins for Fiji written in Java. The Mars source code is available under the BSD-2 license and deposited in several repositories on Github (see "Code availability" section). Videos were analyzed using Mars as follows. First, a discoidal averaging filter was applied to the images to amplify single beads over background. Second, all beads were detected using an intensity threshold followed by local maximum determination. Third, bead positions were determined with subpixel accuracy by 2D Gaussian fitting. Finally, bead positions were linked from frame to frame to reveal tracks using a local radius search on a KDTree for optimal performance (ImgLib2)[24]. These individual steps are combined in a single Peak Tracker command located in the Image package of mars-core.

Tracking results were stored in the Molecule Archive format of Mars together with image metadata, which were saved in either plain text JSON format or with Smile encoding to reduce the file sizes. The coordinates for each bead were stored in molecule records together with parameters (i.e., mean-squared displacement, reversal step, etc.), tags, and segments generated using kinetic change point analysis. All molecule records were given Universally Unique IDs (UUID) for reproducible merging of datasets retaining full processing history (further facilitated by logging of all operations within the Molecule Archives). The results of one FMT experiment yielded Molecule Archives that were 10–20 GB in size. Subsequent merging of multiple archives increased the storage requirements further beyond the memory of normal desktop computers. Therefore, Mars was developed to place

molecule records in virtual storage and retrieve them as needed. Multithreaded processing of these very large datasets was further facilitated by the molecule object core data model, which allows for processing of molecule records in parallel, leveraging the concurrent data structures provided in Java.

A powerful, scriptable API and graphical user interface, allowed for the development of secondary classification algorithms to identify beads with desired qualities (i.e., coilable, showing gyrase activity, etc.) (http://github.com/duderstadt-lab/fmt-scripts). Plots for figures were similarly made leveraging the Mars API but with ImageJ running in Jupyter notebooks written in Groovy or Python. Additional plotting was performed using Prism 8.

**Spatial resolution**. Several distinct factors influence the spatial resolution obtained in our experiments. The magnification, pixel dimensions, and intensity of the bead together result in a fundamental limit in spatial resolution independent of the experimental condition. This can be calculated with the following expression:

$$\sigma = \sqrt{\frac{s^2}{N} + \frac{a^2}{12N} + \frac{8\pi s^4 b^2}{a^2 N^2}},\tag{1}$$

where $N$ is the number of photons, $a$ is the pixel size, $b$ is the standard deviation of the background, and $s_x$ is the standard deviation of the fit of the bead[36]. For MyOne beads with a diameter of 1 µm that are nonspecifically stuck on the surface, we measured $N = 50{,}200$ photons, $a = 1.57$ µm, $b = 107$, $s_x = 0.78$ µm. These measurements were performed using the telecentric lens having ×7 magnification and NA 0.23. The camera was set to 2× binning and collection was performed at 4 Hz. The theoretical limit of spatial resolution for these conditions is ~6 nm. However, several other factors prevent realization of this limit. The most significant of these is mechanical drift. The lens used to achieve the improvement in throughput is very large and tall requiring a custom microscope, which suffers from mechanical drift over time. The standard deviation of the position of beads nonspecifically stuck on the surface after correcting for drift averaging over a 5 min window is 21 nm. The resolution obtained when tracking length changes using mobile beads depends on the DNA length and applied force used in individual experiments (The s.d. of position for mobile beads in gyrase reactions was 81 nm for $n = 29{,}892$, at 33 Hz, for a 1 min time window).

**Force determination**. The dramatic improvement in throughput achieved using FMT relies on the addition of a lateral flow force so that length changes can be observed as lateral changes in position. This removes the requirement for high numerical aperture optics needed for tracking axial position changes. In magnetic tweezers, the force can be calculated using the equipartition theorem given by $F = k_B T l / \langle\delta y^2\rangle$, where $k_B$ is the Boltzmann constant, $T$ is the absolute temperature, $l$ is the length of the molecule, and $\langle\delta y^2\rangle$ is mean-squared displacement of the beads along an axis orthogonal to the force axis, for FMT experiments this is perpendicular to flow (Fig. 1a). However, in FMT experiments only the projected length is observed and not the absolute length, $l$, from the equipartition theorem. We therefore numerically solved for both the length and the force by assuming Worm-Like-Chain (WLC) behavior[37,38] using the set of equations:

$$\delta y^2 = \frac{k_B T l}{F},\tag{2}$$

$$\frac{FP}{k_B T} = \frac{1}{4}\left(1 - \frac{l}{l_0}\right)^{-2} - \frac{1}{4} + \frac{l}{l_0},\tag{3}$$

where $F$ is the force, $P$ is the persistence length (46 nm)[18], $l_0$ is the contour length of the DNA (21 kb, 6.8 µm), $l$ is the extension length of DNA, and the remaining constants are specified above ($T = 296$ K, $k_B = 1.38 \times 10^{-23}$ m² kg s⁻¹ K⁻¹). A utility function was written in the Mars plugins to perform the numerical solving (calculate force and length in the util package in the MarsMath class of mars-core). The bracketing Nth order BrentSolver of the apache commons math3 package was used for the numerical solving. The settings used can be found in the Force Calculator class in the util package of mars-core. Formula 3 is an approximation for the WLC that exhibits 5–7% error at mid extension. If extensive experiments will be conducted within this range an alternative formula, with additional corrections, should be used[39].

To determine the applied force for both single beads and clusters of beads (both were observed in individual experiments), force was calculated individually for each molecule. During the initial stage of each experiment the frame rate was increased from 4 to 33 Hz. Force was calculated at the higher frame rate using the variance of bead motion in the direction perpendicular to flow. Thirty-three hertz was the highest rate achievable for the full sensor of our camera with 2× binning. These measurements had to be conducted with the full sensor to capture all molecules. While considerably higher than the experimental frame rate, 33 Hz is too slow to capture the full extent of bead fluctuations that occur on shorter timescales. Fortunately, this is a well-known phenomenon and a correction factor can be calculated for motion blur due to under sampling[38].

The force versus flow rate results presented in Fig. 3 were corrected for the influence of motion blur using the approach outlined by Wong and Halvorsen[21]

using the correction factor $S$.

$$S = \frac{2}{\alpha} - \frac{2}{\alpha^2}\left(1 - e^{-\alpha}\right), \quad (4)$$

where $\alpha$ is the ratio of the camera integration time $W$ to the trap relaxation time $\tau$, $k$ is the spring constant of the trap (equivalent to force $F$ over length $l$)[40], and $\gamma$ is the friction factor of the particle given by the Stokes' law.

$$\alpha = \frac{W}{\tau} = \frac{kW}{\gamma} = \frac{FW}{l6\pi\eta R}, \quad (5)$$

where $R$ is the radius of the bead and $\eta$ is the viscosity. The corrected variance is the measured variance divided by the correction factor $S$.

$$\langle \delta y^2 \rangle_c = \frac{\delta y^2}{S(\alpha)}. \quad (6)$$

The correction factor $S$ depends on force and length, which are numerically determined using Eqs. 2, 3. Therefore, the correction factor was incorporated into Eq. 1 in a second numerical solver class called Motion Blur Force Calculator in the util package of mars-core. This solver was used to calculate the points in Fig. 3c. The numerically determined correction factor ranged from 2% at lowest flowrate of 2.5 μl/min (used for all gyrase reactions) to 40% at the highest flowrate of 80 μl/min. The blur correction was only applied in Fig. 3. The other figures present experiments conducted at the lowest flow rate of 2.5 μl/min, where motion blur does not significantly influence force estimation.

The magnets were kept at a constant height above the flow cell throughout all experiments. To determine the magnetic force the fluctuations in bead position at zero flow were measured. Under these conditions the numerically determined force is $0.08 \pm 0.04$ pN ($n = 917$ molecules from one experiment). Given an estimate for the bead height above the surface, the force due to drag can be calculated using Stokes' law. However, the low magnification lens used to obtain very high throughput does not allow for direct observations of bead height above the surface. Nevertheless, the height can be estimated using a combination of experimentally determined and calculated values. The attachment site of all DNA molecules is determined using the reversal step at the beginning of all experiments. The projected length can be calculated throughout each experiment. The fluctuations in bead position is perpendicular to flow were used to numerically solve for the force and extension of each DNA molecule. Using these two values the angle and height above the surface can be calculated for the bead position using the following formulas:

$$\theta = \cos^{-1}\frac{l_{xy}}{l} \quad \text{and} \quad z = l\,\sin\theta, \quad (7)$$

where $l_{xy}$ is the projected length in the $xy$-plane, $l$ is the end-to-end extension of the DNA and $\theta$ is the angle with the surface. All gyrase experiments were performed at a flowrate of 2.5 μl/min. Under these conditions the numerically determined force and extension length are $0.2 \pm 0.1$ pN and $4.4 \pm 0.1$ μm, respectively. The projected length is $3.8 \pm 0.1$ μm and we calculate a mean angle of $25° \pm 2°$ and height above the surface (in the $z$ direction) of $2.1 \pm 0.1$ μm (calculated from $n = 917$ molecules from one experiment). The drag force can be calculated using Stokes' law.

$$F_d = 6\,\pi\eta R\nu \quad \text{where} \quad \nu = 2\,\nu_{max}\frac{z}{h}\left(1 - \frac{z}{h}\right) \quad \text{and} \quad \nu_{max} = \frac{3Q}{2wh}, \quad (8)$$

where $F_d$ is the drag force for viscosity $\eta$, bead radius $R$ and flow velocity $\nu$. For a laminar flow the flow velocity is a function of the distance above the surface as given by the central expression, where $h$ is the height of the flow cell and $z$ is the distance of the bead above the surface. And finally, $\nu_{max}$ is the highest flow velocity observed at the center of the flow lane, which can be calculated from the height $h$, width $w$, and the volumetric flow $Q$. The drag force can be calculated using the height above the surface from our experimental data ($z = 2.1$ μm). All the other parameters are known for our system to have the following values: $w = 3$ mm. $h = 100$ μm, $\eta = 8.9 \times 10^{-4}$ Pa*s (for water), $Q = 4.2 \times 10^{-11}$ m³/s. Using these values the drag force is 0.07 pN.

The above calculation is an approximation for the primary experimental condition used for all gyrase reactions reported. The force as a function of flowrate plot reported in Fig. 3c was generated using a range of flowrates. Given that the magnet height was held constant throughout these experiments, the approximation above will not be accurate for the higher flowrates. We would expect the bead height above the surface to decrease to as little as 100 nm at the highest flowrate of 80 μl/min. Surface effects may be observed under these conditions. By lowering the magnet closer to the flow cell at higher flowrates a constant bead height above the surface could be maintained. In the current study, this was not necessary because all gyrase experiments were conducted at very low flow rate and low force where the approximation above holds.

**Molecule classification**. A rigorous set of criteria were used to evaluate and further classify individual molecules generated using the Mars Peak Tracker. First, the mobility of each bead was checked using the mean-squared displacement as well as flow reversal steps at the beginning and end of each experiment. The center point between the two extreme positions during flow reversal was taken as the attachment site to determine the projected length. Next, molecules were checked for coilability (to ensure they were not nicked) by performing a series of clockwise and counterclockwise rotations of the magnets. The standard coiling series was 150

clockwise turns and 150 counterclockwise turns, followed by 150 counterclockwise turns and 150 clockwise turns. To determine if beads were single or multiply tethered, the coiling steps were conducted at high force. For this condition single tethers only showed compaction for positive supercoiling, whereas multiply-tethered beads showed compaction for both coiling steps. Multiply-tethered molecules were then removed based on the slope in the negative coiling region. For gyrase experiments the final uncoiling step was stopped after 40 turns leaving the DNA positively supercoiled in preparation for gyrase arrival. An additional high frame rate (33 Hz) region was added for molecule-by-molecule force determination (using the approach outlined in the force determination section). For gyrase experiments the rate of length change observed during magnet rotations conducted at the experimental flowrate was used to calculate the number of cycles/s.

Bursts of gyrase activity within the subset of coilable molecules were classified using the overall length change from before gyrase arrival and after all supercoiling activity. For $\alpha$ reactions, where the DNA is negatively supercoiled after relaxation, this length difference is negative. For $\chi$ reactions, where the DNA is only relaxed and high force prevents introduction of negative supercoiling, this length difference is positive. The locations of gyrase activity within these selected molecules were identified using a sliding window approach. A window of 12.5 s was used in the region prior to max length to identify the region of positive supercoil relaxation (searching for the highest slope). A window of 25 s was used to identify the region of negative supercoiling (searching for the lowest slope). The difference in window size was due to differences in the duration of the distinct activities. The maximum slopes detected are represented as circles in the scatter plot displayed in Fig. 4d.

In all cases, the Mars API was used to develop scripts for automated secondary classification either using the scripting editor of Fiji or Jupyter notebooks. Plots were created in Jupyter notebooks using matplotlib, Matlab and Prism 8.

**Statistics and reproducibility**. The number of molecules and independent experiments for each observation are reported in the text and figure legends, where space allows. The remaining observation numbers are reported here. The number of positive relaxation ($n_p$) and negative introduction ($n_n$) bursts used to generate Fig. 5b, c were as follows: no drug ($n_p = 1662$, $n_n = 1067$), 78 nM ($n_p = 1492$, $n_n = 1014$), 156 nM ($n_p = 1503$, $n_n = 923$), 312 nM ($n_p = 1761$, $n_n = 942$), 1250 nM ($n_p = 3202$, $n_n = 2050$), 2500 nM ($n_p = 1072$, $n_n = 501$), 5 μM ($n_p = 2647$, $n_n = 1340$), 20 μM ($n_p = 3204$, $n_n = 2609$), and 50 μM ($n = 506$). Each represents molecules observed in single experiments. All key findings reported were confirmed in more than three independent experiments.

**Reporting summary**. Further information on research design is available in the Nature Research Reporting Summary linked to this article.

## Data availability

Raw videos for reproduction of the essential results are available at Zenodo with the following titles and links: Flow Magnetic Tweezers: Gyrase dynamics in absence of drug ciprofloxacin (https://doi.org/10.5281/zenodo.3981513), Flow Magnetic Tweezers: Gyrase dynamics in presence of 20 μM drug ciprofloxacin (https://doi.org/10.5281/zenodo.3981123), Flow Magnetic Tweezers: Gyrase dynamics under 3 different external torque conditions. Part 1/3 (https://doi.org/10.5281/zenodo.3981531), Flow Magnetic Tweezers: Gyrase dynamics under three different external torque conditions. Part 2/3 (https://doi.org/10.5281/zenodo.3981542), Flow Magnetic Tweezers: Gyrase dynamics under three different external torque conditions. Part 3/3 (https://doi.org/10.5281/zenodo.3981545). Additional data and instructions available upon request.

## Code availability

The analysis software used for this work is open source and publicly available in several repositories on Github at https://github.com/duderstadt-lab. The core data model implementation and plugins used for data processing and storage are available at http://github.com/duderstadt-lab/mars-core and the graphical user interface used for classification and plotting is available at http://github.com/duderstadt-lab/mars-fx. The plugin used for length conversion from pixels to microns is available at http://github.com/duderstadt-lab/mars-fmt. Scripts used for region definitions and automated feature classification can be found at http://github.com/duderstadt-lab/fmt-scripts. To facilitate the general use of our software, we have created an ImageJ update site at http://sites.imagej.net/Mars/ and a documentation site http://github.com/duderstadt-lab/mars-docs. The plugins can very easily be installed by adding the update site to any local copy of Fiji. The use of specific commands is described in detail on the documentation site.

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

## Acknowledgements
We would like to thank James M. Berger for the gift of E. coli DNA gyrase. We would also like to thank James M. Berger, Jan Lipfert, and Ralf Jungmann for many insightful discussions. We would like to thank Marta Urbanska and Stefan Diez for suggesting a numerical approximation to obtain forces in the absence of direct length measurements and many helpful discussions. We would like to thank Chris Duderstadt for help in drawing the mechanical specifications for custom machined microscope components. This work was funded by the Deutsche Forschungsgemeinshaft (DFG, German Research Foundation)—SFB863—11166240, a starting grant from the European Research Council (ERC-StG-804098 ReplisomeBypass), and the Max Planck Society. Open Access funding provided by Projekt DEAL

## Author contributions
R.A. and K.E.D. designed the experiments, conducted the analysis and wrote the paper. R.A. performed the experiments.

## Funding

## Competing interests
The authors declare no competing interests.
