## [Peer Review File · Nature Communications]

REVIEWER COMMENTS

Reviewer #1 (Remarks to the Author):

In their manuscript "Flow magnetic tweezers: imaging rare enzymatic events with single molecule precision" Rohit Agarwal and Karl Duderstadt describe a massively parallel single-molecule manipulation instrument that combines magnetic tweezers and flow stretching approaches in an innovative manner. The instrument is relatively simple in design, which is a feature. It consists of a flow chamber above which is mounted a pair of magnets in a conventional magnetic tweezers configuration. The flow cell is illuminated from the side and imaged through a low-magnification telecentric imaging lens assembly onto a 20 megapixel camera, which permits imaging an area of several mm on a side. Long ~22kb DNA molecules tether 1 micron magnetic particles to the surface of the flowcell. Force is mainly controlled through the application of a controlled flow through the flowcell, which combines with the small vertical force applied by the magnets to produce a net stretching force. The advantage of this configuration is that displacement is largely in the plane of the microscope image and can be measured via conventional particle tracking approaches. The heart of the approach is the software suite that controls the flow, the rotation of the magnets, and most importantly, determines the applied force and DNA end to end extension by simultaneously applying equipartition to the variance of the bead position perpendicular to the flow, and the wormlike chain model to the extension of the DNA, which must be inferred due to the fact that the measured extension is a projection of the actual extension into the plane. The system is calibrated with varying flow and the force-extension relation for DNA is measured along with the force at which supercoiled DNA buckles are obtained as control measures of the force calibration. This approach permits the tracking of ~ 50 000 particles, which the software sorts out based on a set of criteria resulting in many thousand tethers that are viable. As a proof of concept the authors track the activities of DNA gyrase in relaxing positive supercoils, introducing negative supercoils, and how it is inhibited by ciprofloxacin. These experiments qualitatively reproduce previous measures of gyrase activity achieving orders of magnitude more measures in a single three hour experiment than from previous single-molecule measurements of gyrase activity. These experiments and the demonstration of rare double strand breakage of the DNA during gyrase relaxation activity highlight the strengths of this approach.

This is a valuable addition to the single-molecule biophysics instrumentation spanning the challenging regime between single-molecule and high throughput approaches. The simplicity of the approach and the availability of the source code will make this approach relatively straightforward for other labs to implement. The concept is solid, the implementation is well-done and the appropriate controls have been done. The proof of concept experiments were well-designed to highlight the strengths of the approach. Overall this is a solid manuscript that is worthy of publication. It will be of interest to a broad community of biophysicists and will likely be implemented due to its simple and elegant design. I have several detailed points that the authors should address before publication of the manuscript as detailed below.

General comments

The authors do a good job of highlighting the strengths of this approach but these strengths come at a cost, and these costs should be spelled out. This is not to detract from the work and the methodology, but any approach has to make trade-offs and I think that a clear discussion of these trade-offs would benefit rather than detract from the work. In this instance the massively parallel aspect of this technique comes at the cost of temporal and spatial resolution at a minimum, and likely other more subtle considerations. For example, the length of the DNA tethers needs to be sufficient so that the timescale of the fluctuations remains half of the camera bandwidth, and the length of the tethers also likely impacts the accuracy of the extension and force calibrations. I would like to see a brief but clear discussion of the strengths and limitations of this approach. Again -this is not to detract from the work but simply to place it in the appropriate context among possible single-molecule approaches. Perhaps this could also include a brief discussion of future improvements in this approach.

Detailed comments:

1. Line 135: "Buckling of underwound DNA, shown in Fig. 2a, b, occurs at ~ 1 pN. Whereas, buckling of overwound DNA occurs at ~ 6.5 pN. To determine the flowrate for each transition, flow

ramp experiments were performed in which flow was reduced or increased gradually (Fig. 3a). For both over and underwound DNA, rapid buckling and extension were observed at distinct flowrates. Transition flowrates, $9.2 \pm 0.1 \mu\text{l}/\text{min}$ (median \pm s. e. median) for underwound DNA and $68.7 \pm 0.7 \mu\text{l}/\text{min}$ (median \pm s. e. median) for overwound DNA". This does not make sense. More context should be given. It depends on the number of turns imposed for positive supercoiling, without this information the statement does not make sense. For negative supercoiling the transition force is more or less constant but for positive it depends on the level of supercoiling, which should be indicated.

2. 148: how do the computed forces compare with calculated forces based on particle size, flow rate, DNA tether length and magnetic force? What is the force precision? what is the standard deviation of forces – in other words what is the uncertainty in force for a given particle? Also it would be good to know what the magnetic force is.

3. For experiments are the forces always calculated from the variance, or was the flow calibration assumed constant?

4. 188: previous reports found a burst mode in gyrase relaxing positively supercoiled DNA. Was this observed? Previous data also obtained force velocity curves for DNA gyrase. How do the velocity versus force curves compare with previous measurements? Can the traces be used to verify individual enzyme activity? This was not described - was there a selection step to identify molecules possibly being acted on by multiple gyrases?

5. 285 –The statistics are impressive but limit the significant figures based on the uncertainty – 98 ± 11 million events would be a better way to present this measure.

6. 295 what was the applied force during the cipro experiments? or the range of applied forces?

7. 300- were there control experiments with no gyrase and sds to look for the release of DNA due to SDS disruption of the tether attachment points? Can the authors measure cipro-induced nicking? Many type II poisons do not induce double stranded breaks but rather single-stranded breaks, can the authors probe this in the data?

8. 316: "several populations of ds breaks" this not clear – what is meant by "several populations"?

9. 450: how was the tracking resolution determined?

10. 474: How sensitive is the force routine to errors and uncertainties in the measured or assumed values? Typically a DNA force extension curve is a useful verification to demonstrate the ability to accurately measure force and extension. Also, what is the expected sensitivity to the DNA length? Some general caveats would be useful – with out detracting from the power of thee technique. The sampling rate of the camera will set limits on the speed of reactions that can be measured and the jointly limit the force and length of molecules that can be calibrated. The effect of the bead on the projection of the DNA along the flow axis will also become increasingly important as the DNA length is decreased. These are just some considerations that should be discussed. This is a powerful highly multiplexed approach but it is not the best option for some class of experiments – all of which would benefit from higher throughput.

11. Fig 1. What is the force applied by the magnets?

12. Fig 2 B what are the error bars on the points in the graph?

13. Figure 3. The buckling for positively supercoiled DNA depends on sigma. This should be included – this is a missing but important control parameter for the positive buckling transition. Negative is constant for different levels of sigma but positive transition depends on the level of positive supercoiling, which should be specified.

14. Fig 4 A the graph showing negative supercoil introduction on the right should show decreasing extension as a function of time not increasing. Can the authors observe individual steps of Linking number? How is there a selection step for individual single gyrase trajectories?

15. How do the force -velocity relations compare with previous measures for DNA gyrase?

16. Fig 5 633 what was the fit function to obtain the inhibition constant IC50?

17. Fig 5 654: was there a control done with DNA only and SDS to check for loss of DNA tethers? in G the "survive " curve is not specified – it also not clear if it and the breakage curve are required since the information is redundant.

18. Supplemental 51 "First, due to engagement as displayed in ii" what is "engagement"? is this gyrase binding? If so do the changes in extension correspond to the expected changes due to gyrase binding and wrapping DNA?

Reviewer #2 (Remarks to the Author):

This is an interesting new setup with a very large throughput for magnetic tweezers. The ability to study the torsional behavior of so many molecules simultaneously is really impressive as well as the possibility to observe rare events which are definitely a challenge in single molecule experiments.

The paper is carefully written and the biological results of high quality, the work is clearly interesting for the community. I recommend publication with a few minor corrections and additions which I think could further improve it.

Having such a high throughput is very interesting but one issue is the force calibration that the authors solve with a nice trick exploiting the well-known elasticity of the dsDNA and the equipartition theorem. The authors should emphasize that the calibration assumes the user is working with dsDNA and if I am correct that the persistence length is 50 nm and is imposed in the process. In their comparison table, it would be nice to have the resolution of the different devices compared.

Some information which I did not find and would be worth providing is what is occurring in the z direction where no direct measurement is possible: giving at least the value of the magnetic force F_m would help, also giving the estimated distance separating the bead from the surface versus the drag force would also be nice (I suspect that the bead is pretty close to the surface, does it touch it? Or is the bead reasonably away from the surface so that no increase of the viscosity occurs?). This effect will also define the limit in force that is possible with this setup which is not really discussed nor the size of the molecule that can be used (short molecules will not be possible). Again explaining these limitations is important for future users.

The point-by-point response to the reviewer comments can be found below in blue after each point. New sections added to the main text are highlighted in blue.

Reviewer #1 (Remarks to the Author):

In their manuscript "Flow magnetic tweezers: imaging rare enzymatic events with single molecule precision" Rohit Agarwal and Karl Duderstadt describe a massively parallel single-molecule manipulation instrument that combines magnetic tweezers and flow stretching approaches in an innovative manner. The instrument is relatively simple in design, which is a feature. It consists of a flow chamber above which is mounted a pair of magnets in a conventional magnetic tweezers configuration. The flow cell is illuminated from the side and imaged through a low-magnification telecentric imaging lens assembly onto a 20 megapixel camera, which permits imaging an area of several mm on a side. Long ~22kb DNA molecules tether 1 micron magnetic particles to the surface of the flowcell. Force is mainly controlled through the application of a controlled flow through the flowcell, which combines with the small vertical force applied by the magnets to produce a net stretching force. The advantage of this configuration is that displacement is largely in the plane of the microscope image and can be measured via conventional particle tracking approaches. The heart of the approach is the software suite that controls the flow, the rotation of the magnets, and most importantly, determines the applied force and DNA end to end extension by simultaneously applying equipartition to the variance of the bead position perpendicular to the flow, and the wormlike chain model to the extension of the DNA, which must be inferred due to the fact that the measured extension is a projection of the actual extension into the plane. The system is calibrated with varying flow and the force-extension relation for DNA is measured along with the force at which supercoiled DNA buckles are obtained as control measures of the force calibration. This approach permits the tracking of ~ 50 000 particles, which the software sorts out based on a set of criteria resulting in many thousand tethers that are viable. As a proof of concept the authors track the activities of DNA gyrase in relaxing positive supercoils, introducing negative supercoils, and how it is inhibited by ciprofloxacin. These experiments qualitatively reproduce previous measures of gyrase activity achieving orders of magnitude more measures in a single three hour experiment than from previous single-molecule measurements of gyrase activity. These experiments and the demonstration of rare double strand breakage of the DNA during gyrase relaxation activity highlight the strengths of this approach.

This is a valuable addition to the single-molecule biophysics instrumentation spanning the challenging regime between single-molecule and high throughput approaches. The simplicity of the approach and the availability of the source code will make this approach relatively straightforward for other labs to implement. The concept is solid, the implementation is well-done and the appropriate controls have been done. The proof of concept experiments were well-designed to highlight the strengths of the approach. Overall this is a solid manuscript that is worthy of publication. It will be of interest to a broad community of biophysicists and will likely be implemented due to its simple and elegant design. I have several detailed points that the authors should address before publication of the manuscript as detailed below.

General comments

The authors do a good job of highlighting the strengths of this approach but these strengths come at a cost, and these costs should be spelled out. This is not to detract from the work and the methodology, but any approach has to make trade-offs and I think that a clear discussion of these trade-offs would benefit rather than detract from the work. In this instance the massively parallel aspect of this technique comes at the cost of temporal and spatial resolution at a minimum, and likely other more subtle considerations. For example, the length of the DNA tethers needs to be sufficient so that the timescale of the fluctuations remains half of the camera bandwidth, and the length of the tethers also likely impacts the accuracy of the extension and force calibrations. I would like to see a brief but clear discussion of the strengths and limitations of this approach. Again -this is not to detract from the work but simply to place it in the appropriate context among possible single-molecule approaches. Perhaps this could also include a brief discussion of future improvements in this approach.

We would like to thank the reviewer for their very positive assessment of our manuscript. We are especially pleased they felt the appropriate controls and proof of concept experiments were performed. Moreover, they appreciated the very rare double-strand break intermediates we were able to visualize using the improved throughput of our method.

We would also like to thank the reviewer for their very thoughtful consideration of our manuscript. They have pointed out several important technical considerations in the implementation of flow magnetic tweezers, which were not fully addressed in our manuscript. We have now addressed these points with additional main text and methods sections. We completely agreed with the reviewer that there are important trade-offs made to achieve very high throughput. These points must be carefully considered in the context of the system and dynamics that are studied. We have now explained these trade-offs with a new paragraph at the end of the first results section.

Before responding to the specific points, we would like to address the spatial resolution and camera bandwidth limitations raised in the general comments.

Spatial resolution

Several distinct factors influence the spatial resolution obtained in our experiments. The magnification, pixel dimensions, and intensity of the bead together results in a fundamental limit in spatial resolution independent of the experimental condition. This can be calculated with the following expression:

$$\sigma = \sqrt{\frac{s^2}{N} + \frac{a^2}{12N} + \frac{8\pi s^4 b^2}{a^2 N^2}}$$

where N is the number of photons, a is the pixel size, b is the standard deviation of the background, and s_x is the standard deviation of the fit of the bead (Thompson, R.E., Larson, D.R. & Webb, W.W. *Biophys. J.* **82**, 2775-2783 (2002).). For MyOne beads with a diameter of 1 μm that are nonspecifically stuck on the surface, we measured $N = 50,200$ photons, $a = 1.57 \mu\text{m}$, $b = 107$, $s_x = 0.78 \mu\text{m}$. These measurements were performed using our telecentric lens having 7x magnification and NA 0.23. The camera was set to 2x binning and collection was performed at 4 Hz. The theoretical limit of spatial resolution for these conditions is ~ 6 nm. However, several other factors prevent realization of this limit. The most significant of these is mechanical drift. The lens used to achieve the improvement in throughput is very large and tall requiring a custom microscope which suffers from mechanical drift overtime. The standard deviation of the position of beads nonspecifically stuck to the surface after correcting for drift is 21 nm based on averaging a 5 minute time window for $N = 321$ molecules. The resolution obtained when tracking length changes using mobile beads depends on the DNA length and applied force used in individual experiments (The s.d. of position for mobile beads in gyrase reactions was 81 nm for $N = 29,892$, at 33 Hz, for a 1 minute time window).

In response to comments by Reviewer #2, the spatial resolution has been added to Table 1. A new spatial resolution section has also been added to the methods section with the above calculation.

We appreciate the reviewer's comment that the range of phenomena that can be studied will be limited by the DNA tether length and ability of the camera to capture the bead motion. These aspects of the experimental design are critical considerations that will influence the experimental outcome. First, we will consider the influence of camera bandwidth on our force estimation.

Camera bandwidth

During the initial stage of each experiment the frame rate was increased from 4 Hz to 33 Hz and force was calculated individually for each molecule. Force was calculated at the higher frame rate using the variance of bead motion in the direction perpendicular to flow. 33 Hz was the highest rate achievable for the full sensor of our camera with 2x binning. These measurements had to be conducted with the full sensor to capture all molecules. While considerably higher than the experimental frame rate, 33 Hz is too slow to capture the full extent of bead fluctuations that occur on shorter timescales. Fortunately, this is a well-known phenomenon and a correction factor can be calculated for motion blur due to undersampling.

The force vs flowrate results presented in **Fig. 3** were corrected for the influence of motion blur using the approach outlined by Wong and Halvorsen³⁵ using the correction factor S .

$$S = \frac{2}{\alpha} - \frac{2}{\alpha^2} (1 - e^{-\alpha})$$

Where α is the ratio of the camera integration time W to the trap relaxation time τ , k is the spring constant of the trap (equivalent to force F over length l^{36}), and γ is the friction factor of the particle given by the Stokes' law.

$$\alpha = \frac{W}{\tau} = \frac{kW}{\gamma} = \frac{FW}{l6\pi\eta R}$$

Where R is the radius of the bead and η is the viscosity. The corrected variance is the measured variance divided by the correction factor S .

$$\langle \delta y^2 \rangle_c = \frac{\langle \delta y^2 \rangle}{S(\alpha)}$$

The correction factor S depends on force and length, which are numerically determined using the equipartition theorem and WLC model. Therefore, the correction factor was incorporated into the equipartition theorem in a second numerical solver class called MotionBlurForceCalculator in the util package of mars-core. This solver was used to calculate the points in **Fig. 3c**. The numerically determined correction factor ranged from 2% at lowest flowrate of 2.5 ul/min (used for all gyrase reactions) to 40% at the highest flowrate of 80 ul/min. The blur correction was only applied in **Fig. 3**. The other figures present experiments conducted at the lowest flowrate of 2.5 ul/min where motion blur does not significantly influence force estimation.

DNA length (duplicated in the response to reviewer #2)

We agree that short DNA molecules would be difficult to study with our method. Ultimately the laminar flow profile will place limitations on the absolute forces that could be applied. For example, according to the WLC model, a DNA with a contour length of 1 kb would have an end-to-end length of 270 nm at 1 pN. If the magnetic force was held fixed at 0.08 pN, as done in our current experiments, the flowrate required to maintain 1 pN of force would be in excess of 1,000 ul/min. This flowrate could not be achieved with our current instrumentation and would require large amounts of any enzymes or reagents needed for the studies. This issue could be partially overcome by increasing the force from the magnet by bringing it closer to the surface. Assuming the magnetic force was equally balanced with the drag force the flowrate required would be closer to 200 ul/min. However, for such a

short DNA, the limited resolution of our low magnification lens might pose further challenges for detecting changes in projected length, which in this case may be on the order of 10s of nms.

As pointed out by the reviewer, this scenario would also result in substantial motion blur using the current data collection parameters and require a large motion blur correction factor. Therefore, work with shorter DNA would require a change in the setup allowing for higher camera bandwidth and smaller fields of view.

Detailed comments:

1. Line 135: "Buckling of underwound DNA, shown in Fig. 2a, b, occurs at ~ 1 pN. Whereas, buckling of overwound DNA occurs at ~ 6.5 pN. To determine the flowrate for each transition, flow ramp experiments were performed in which flow was reduced or increased gradually (Fig. 3a). For both over and underwound DNA, rapid buckling and extension were observed at distinct flowrates. Transition flowrates, 9.2 ± 0.1 $\mu\text{l}/\text{min}$ (median \pm s. e. median) for underwound DNA and 68.7 ± 0.7 $\mu\text{l}/\text{min}$ (median \pm s. e. median) for overwound DNA". This does not make sense. More context should be given. It depends on the number of turns imposed for positive supercoiling, without this information the statement does not make sense. For negative supercoiling the transition force is more or less constant but for positive it depends on the level of supercoiling, which should be indicated.

We are pleased that the reviewer noticed this critical information was missing. The positive buckling transitions were evaluated at a sigma of 0.075 which corresponded to 150 turns on average in each molecule. We have added these details to the text at the section indicated.

2. 148: how do the computed forces compare with calculated forces based on particle size, flow rate, DNA tether length and magnetic force? What is the force precision? What is the standard deviation of forces – in other words what is the uncertainty in force for a given particle? Also it would be good to know what the magnetic force is.

(duplicated in the response to reviewer #2)

Thank you for this comment. To perform this calculation we must first determine the bead height above the surface. The low magnification lens we used to obtain very high throughput does not allow us to directly observe the height of the bead above the surface. However, we can make an estimate of the height in the z direction using a combination of experimentally determined and calculated values. We determine the attachment site of all DNA molecules using the reversal step at the beginning of all experiments. This allows us to monitor the projected length throughout our experiments. Furthermore, we have used the fluctuations in bead position perpendicular to flow to numerically solve for the force and extension of each DNA molecule. Using these two values we can calculate the angle and height above the surface for the bead position using the following formulas.

$$\theta = \cos^{-1} \frac{l_{xy}}{l} \quad \text{and} \quad z = l \sin \theta$$

Where l_{xy} is the projected length in the xy-plane, l is the end-to-end extension of the DNA and θ is the angle with the surface. All gyrase experiments were performed at a flowrate of 2.5 $\mu\text{l}/\text{min}$. Under these conditions the numerically determined force and extension length are 0.2 ± 0.1 pN and 4.4 ± 0.1 μm , respectively. The projected length is 3.8 ± 0.1 μm and we calculate a mean angle of $25^\circ \pm 2^\circ$ and height above the surface (in the z direction) of 2.1 ± 0.1 μm (calculated from $N = 917$ molecules).

To determine the magnetic force we have measured the fluctuations in bead position at zero flow. Under these conditions the numerically determined force is 0.08 ± 0.04 pN. The drag force can be calculated using Stokes' law.

$$F_d = 6 \pi \eta R v \quad \text{where} \quad v = 2 v_{max} \frac{z}{h} \left(1 - \frac{z}{h}\right) \quad \text{and} \quad v_{max} = \frac{3Q}{2wh}$$

Where F_d is the drag force for viscosity η , bead radius R and flow velocity v . For a laminar flow the flow velocity is a function of the distance above the surface as given by the central expression where h is the height of the flow cell and z is the distance of the bead above the surface. And finally, v_{max} is the highest flow velocity observed at the center of the flow lane, which can be calculated from the height h , width w , and the volumetric flow Q . The drag force can be calculated using the height above the surface from our experimental data ($z = 2.1$ μm , from above). All the other parameters are known for our system to have the following values: $w = 3$ mm, $h = 100$ μm , $\eta = 8.9 \times 10^{-4}$ Pa*s (for water), $Q = 4.2 \times 10^{-11}$ m³/s. Using these values the drag force is 0.07 pN. Adding the forces together yields a total calculated force of 0.17 ± 0.04 pN, which is within error of our measured force of 0.2 ± 0.1 pN presented in Figure 3 (with bead clusters excluded from analysis).

The above calculation is an approximation for the primary experimental condition used for all gyrase reactions reported. The force as a function of flowrate plot reported in Figure 3c was generated using a range of flowrates. Given that the magnet height was held constant throughout these experiments, the approximation above will not be accurate for the higher flowrates. We would expect the bead height above the surface to decrease to as little as 100 nm at our highest flowrate of 80 $\mu\text{l}/\text{min}$. Surface effects may be observed under these conditions. By lowering the magnet closer to the flow cell at higher flowrates a constant bead height above the surface could be maintained. In the current study this was not necessary because all gyrase experiments were conducted at very low flowrate and force where the approximation above holds.

Highlighted as a feature in the manuscript, both single beads and bead clusters were observed. This resulted in a broad distribution of force within single experiments (0.6 ± 0.5 pN, $N = 29,892$ molecules, for a typical gyrase reaction). Therefore, molecule-by-molecule force calibration was performed to ensure the correct force was determined for each individual molecule. Importantly, conditions favoring single beads were used for the force calibration results presented in Figure 3 and bead clusters were excluded.

3. For experiments are the forces always calculated from the variance, or was the flow calibration assumed constant?

The force was calculated directly from the variance in all experiments. This was done individually for all molecules. As detailed in the text, bead clumps attached to the end of some DNA molecules. These individual molecules then had higher applied force. To account for these differences and leverage them to obtain force dependent information at fixed flow rate and magnet position, we individually calculated the force for each molecule during each experiment using a higher frame rate region prior to introduction of enzyme (as outline in our general response). We have reviewed the main text to ensure this point is highlighted clearly.

4. 188: previous reports found a burst mode in gyrase relaxing positively supercoiled DNA. Was this observed? Previous data also obtained force velocity curves for DNA gyrase. How do the velocity versus force curves compare with previous measurements? Can the traces be used to verify individual enzyme activity? This was not described - was there a selection step to identify molecules possibly being acted on by multiple gyrases?

Yes, we observe steps and bursts in gyrase activity during positive supercoil relaxation (see example traces below). All gyrase reactions were conducted at 0.2 nM of enzyme to favor the observation of single enzyme events. Moreover, the start times of gyrase activity are distributed over a 2-minute time period consistent with single enzyme activity. No additional filter was applied to specifically locate multi-enzyme events. Interestingly, DNA molecules remained extended during treadmill assays conducted at rates up to 8 rotations/s (Figure 5f). The peak velocity of a single enzyme would not be sufficient to removed supercoil introduction at this rate. This suggests multiple enzymes per molecule for this condition. But the formation of positive supercoils in the molecule is also expected to enhance binding, so we can't extrapolate from these conditions to make a statement about the mean number of gyrases for our normal reaction condition. This observation has been mentioned in the main text.

The trend observed in the velocity vs force curves displayed in Figure 4c qualitatively agrees with the results of Nöllman *et al.* NSMB 2007 (see image below, take note of the difference in x ranges). The peak value of ~ 1.15 cycles for positive supercoil relaxation in our manuscript is within error of the peak value reported in Figure 7a of Nöllman *et al.* NSMB 2007. In both cases, the velocity decreases to a plateau below 1 cycle/s beyond 1 pN of force. Negative supercoiling is only observed at very low force below 0.5 pN with a rapid drop in activity prior to that cutoff. It is worth noting that velocities lower than the peak value were observed in Nöllman *et al.* at very low forces (~ 0.1 pN). We did not observe this but all our observations were conducted above 0.2 pN of force. Our peak velocity also quantitatively agrees with Basu *et al.* NSMB 2012. We have mentioned in the text the agreement of our observations with these previous works.

Figure 7a of Nöllmann et al. reprinted by permission from Springer Nature: Nöllmann, M., Stone, M., Bryant, Z. et al. Multiple modes of *Escherichia coli* DNA gyrase activity revealed by force and torque. *Nat Struct Mol Biol* **14**, 264–271 (2007). <https://doi.org/10.1038/nsmb1213>

5. 285 –The statistics are impressive but limit the significant figures based on the uncertainty – 98 +/-11 million events would be a better way to present this measure.

We would like to thank the reviewer for pointing out this mistake. The manuscript has been corrected.

6. 295 what was the applied force during the cipro experiments? or the range of applied forces?

The flowrate and magnet height remained constant during the cipro experiments to match the conditions for the normal gyrase reactions. The forces on each individual molecule depends on whether a single bead or a bead cluster is attached at the end. Molecule-by-molecule force determination was used to account for these differences. The forces for the cipro experiments were typically 0.6 ± 0.5 pN as determined from $N = 29,892$ molecules.

7. 300- were there control experiments with no gyrase and sds to look for the release of DNA due to SDS disruption of the tether attachment points? Can the authors measure cipro-induced nicking? Many type II poisons do not induce double stranded breaks but rather single-stranded breaks, can the authors probe this in the data?

Yes, several control experiments were conducted with SDS washes and no gyrase. Tethers were not lost due to SDS washing. To address the question of whether cipro induces nicking, we manually evaluated hundreds of molecules for conditions in the presence and absence of cipro. Based on this examination we do not see an increase in nicking in the presence of cipro over background. In both conditions nicking is very rare and only 2-3 were observed in this manual comparison. We think this is background nicking that is not specific to cipro.

8. 316: “several populations of ds breaks” this not clear – what is meant by “several populations”?

Most dsDNA breaks were observed during SDS treatment after gyrase reactions conducted with cipro. The second population were rare dsDNA breaks observed during normal gyrase activity. This line has been reworded to prevent confusion. We thank the reviewer for pointing this out.

9. 450: how was the tracking resolution determined?

We have added a new methods section discussing the spatial resolution limitations of FMT. This was also addressed in the general comments above. In short, stuck beads were tracked and drift corrected for a 5 minute time window. Then the standard deviation in bead position was 21 nm using this method. We used this approach to allow for comparison with the other methods outlined in Table 1, which used the same approach to calculate the spatial resolution.

10. 474: How sensitive is the force routine to errors and uncertainties in the measured or assumed values? Typically a DNA force extension curve is a useful verification to demonstrate the ability to accurately measure force and extension. Also, what is the expected sensitivity to the DNA length? Some general caveats would be useful – with out detracting from the power of thee technique. The sampling rate of the camera will set limits on the speed of reactions that can be measured and the jointly limit the force and length of molecules that can be calibrated. The effect of the bead on the projection of the DNA along the flow axis will also become increasingly

important as the DNA length is decreased. These are just some considerations that should be discussed. This is a powerful highly multiplexed approach but it is not the best option for some class of experiments – all of which would benefit from higher throughput.

We thank the reviewer for raising these important points. These points have been addressed in our general response where we calculated and corrected for inaccuracies in our force calibration method due to our limited camera bandwidth. We also discussed the difficulties in studying short DNAs.

We have added a new paragraph at the end of the first results section outlining these issues.

11. Fig 1. What is the force applied by the magnets?

To determine the magnetic force we have measured the fluctuations in bead position at zero flow. Under these conditions the numerically determined force is 0.08 ± 0.04 pN ($N = 917$ molecules). This estimate has been added to the force calibration section in the methods.

12. Fig 2 B what are the error bars on the points in the graph?

The length vs sigma dependence plotted here as a function of force was calculated from a single molecule and is displayed to illustrate that using flow as the primary means to apply force results in the same trends qualitatively as observed in previous studies using magnetic tweezers. We have corrected the figure legend for clarification.

13. Figure 3. The buckling for positively supercoiled DNA depends on sigma. This should be included – this is a missing but important control parameter for the positive buckling transition. Negative is constant for different levels of sigma but positive transition depends on the level of positive supercoiling, which should be specified.

We would like to thank the reviewer for raising this important point. We used 150 turns and a sigma of 0.075 for the positive buckling experiments. This has been added in the text and to the legend of figure 3.

14. Fig 4 A the graph showing negative supercoil introduction on the right should show decreasing extension as a function of time not increasing. Can the authors observe individual steps of Linking number? How is there a selection step for individual single gyrase trajectories?

In Fig 4a the graphs were showing the total number of cycles based on the slope for coiling determined at the beginning of each experiment. To avoid confusion we have now changed to length where positive relaxation leads to extension and negative introduction leads to compaction. No additional selection step was added for individual gyrase trajectories (further discussion can be found in the response to comment 4). We observe steps in gyrase activity for some molecules (see comment 4). The length change associated with most of these steps is more than one gyrase cycle, which should be ~ 120 nm under our conditions. However, we are developing reaction conditions that should further favor his stepping behavior in the context of another manuscript.

15. How do the force -velocity relations compare with previous measures for DNA gyrase?

This point is discussed in response to comment #4 above.

16. Fig 5 633 what was the fit function to obtain the inhibition constant IC50?

The IC50 was determined using an inhibitor vs response equation provided in Prism 8 ($Y = \text{Bottom} + (\text{Top} - \text{Bottom}) / (1 + (X / \text{IC50}))$). This has been clarified in the figure legend.

17. Fig 5 654: was there a control done with DNA only and SDS to check for loss of DNA tethers?

Yes, several control experiments were performed with SDS washes in the absence of gyrase and tethers were not lost. This has now been mentioned in the main text.

in G the “survive “ curve is not specified – it also not clear if it and the breakage curve are required since the information is redundant.

We completely agree that the “survive” curve is not clearly defined and redundant. We have removed it from the figure.

18. Supplemental 51 “First, due to engagement as displayed in ii” what is “engagement”? is this gyrase binding? If so do the changes in extension correspond to the expected changes due to gyrase binding and wrapping DNA?

Yes engagement means binding here (this has been clarified in the supplement). Yes, we believe the changes are roughly what would be expected due to gyrase binding and wrapping. We are investigating this question in more detail in the context of another manuscript.

Reviewer #2 (Remarks to the Author):

This is an interesting new setup with a very large throughput for magnetic tweezers. The ability to study the torsional behavior of so many molecules simultaneously is really impressive as well as the possibility to observe rare events which are definitely a challenge in single molecule experiments.

The paper is carefully written and the biological results of high quality, the work is clearly interesting for the community. I recommend publication with a few minor corrections and additions which I think could further improve it.

We are very happy the reviewer found our manuscript impressive, of high quality, and believes it will be of broad interest to the community. The reviewer has raised several interesting and important points about the technical implementation of our approach and possible unavoidable limitations resulting from the low magnification used to achieve very high throughput.

We have addressed these comments point-by-point below by incorporating additional experimental details and discussion in the revised manuscript:

Having such a high throughput is very interesting but one issue is the force calibration that the authors solve with a nice trick exploiting the well-known elasticity of the dsDNA and the equipartition theorem. The authors should emphasize that the calibration assumes the user is working with dsDNA and if I am correct that the persistence length is 50 nm and is imposed in the process.

We agreed and have now emphasized in the third paragraph of the first results section that our calibration depends on working with dsDNA. We used a constant value of 46 nm for the persistence length in the WLC for all force calculations as recommended by Lipfert *et al.* PNAS 2014. Our software and scripts deposited on Github allow users to change this parameter if their experimental conditions require it.

We are very glad the reviewer noticed the persistence length was not mentioned in the text. We have now added to the manuscript the values for all constants used for force calibration (third paragraph of the first results section and in the Force determination methods section).

In their comparison table, it would be nice to have the resolution of the different devices compared.

This is an excellent suggestion. We have added the reported resolution estimates for all methods to the comparison table.

Some information which I did not find and would be worth providing is what is occurring in the z direction where no direct measurement is possible: giving at least the value of the magnetic force F_m would help, also giving the estimated distance separating the bead from the surface versus the drag force would also be nice (I suspect that the bead is pretty close to the surface, does it touch it? Or is the bead reasonably away from the surface so that no increase of the viscosity occurs?).

Thank you for this comment. As you point out, the low magnification lens we used to obtain very high throughput does not allow us to directly observe the height of the bead above the surface. However, we can make an estimate of the height in the z direction using a combination of experimentally determined and calculated values. We determine the attachment site of all DNA molecules using the reversal step at the beginning of all experiments. This allows us to monitor the projected length throughout our experiments. Furthermore, we have used the fluctuations in bead position perpendicular to flow to numerically solve for the force and extension of each DNA molecule. Using these two values we can calculate the angle and height above the surface for the bead position using the following formulas.

$$\theta = \cos^{-1} \frac{l_{xy}}{l} \quad \text{and} \quad z = l \sin \theta$$

Where l_{xy} is the projected length in the xy-plane, l is the end-to-end extension of the DNA and θ is the angle with the surface. All gyrase experiments were performed at a flowrate of 2.5 $\mu\text{l}/\text{min}$. Under these conditions the numerically determined force and extension length are 0.2 ± 0.1 pN and 4.4 ± 0.1 μm , respectively. The projected length is 3.8 ± 0.1 μm and we calculate a mean angle of $25^\circ \pm 2^\circ$ and height above the surface (in the z direction) of 2.1 ± 0.1 μm (calculated from $N = 917$ molecules). We do not expect any surface interactions or changes in viscosity at this distance from the surface.

To determine the magnetic force we have measured the fluctuations in bead position at zero flow. Under these conditions the numerically determined force is 0.08 ± 0.04 pN. The drag force can be calculated using Stokes' law.

$$F_d = 6 \pi \eta R v \quad \text{where} \quad v = 2 v_{max} \frac{z}{h} \left(1 - \frac{z}{h}\right) \quad \text{and} \quad v_{max} = \frac{3Q}{2wh}$$

Where F_d is the drag force for viscosity η , bead radius R and flow velocity v . For a laminar flow the flow velocity is a function of the distance above the surface as given by the central expression where h is the height of the flow cell and z is the distance of the bead above the surface. And finally, v_{max} is the highest flow velocity observed at the center of the flow lane, which can be calculated from the height h , width w , and the volumetric flow Q . The drag force can be calculated using the height above the surface from our experimental data ($z = 2.1$ μm). All the other parameters are known for our system to have the following values: $w = 3$ mm, $h = 100$ μm , $\eta = 8.9 \times 10^{-4}$ Pa*s (for water), $Q = 4.2 \times 10^{-11}$ m^3/s . Using these values the drag force is 0.07 pN. Adding the forces together yields a total calculated force of 0.17 ± 0.04 pN, which is within error of our measured force of 0.2 ± 0.1 pN.

The above calculation is an approximation for the primary experimental condition used for all gyrase reactions reported. The force as a function of flowrate plot reported in Figure 3c was generated using a range of flowrates. Given that the magnet height was held constant throughout these experiments, the approximation above will not be accurate for the higher flowrates. We would expect the bead height above the surface to decrease to as little as 100 nm at our highest flowrate of 80 $\mu\text{l}/\text{min}$. Surface effects may be observed under these conditions. By lowering the magnet closer to the flow cell at higher flowrates a constant bead height above the surface could be maintained. In the current study this was not necessary because all gyrase experiments were conducted at very low flowrate and force where the approximation above holds.

We have added these calculations to the force determination section.

This effect will also define the limit in force that is possible with this setup which is not really discussed nor the size of the molecule that can be used (short molecules will not be possible). Again explaining these limitations is important for future users.

We agree that short DNA molecules would be difficult to study with our method. Ultimately the laminar flow profile will place limitations on the absolute forces that could be applied. For example, according to the WLC model, a DNA with a contour length of 1 kb would have an end-to-end length of 270 nm at 1 pN. If the magnetic force was held fixed at 0.08, as done in our current experiments, the flowrate required to maintain 1 pN of force would be in excess of 1,000 $\mu\text{l}/\text{min}$. This flowrate could not be achieved with our current instrumentation and would require large amounts of any enzymes or reagents needed for the studies. This issue could be partially overcome by increasing the force from the magnet by bringing it closer to the surface. Assuming the magnetic force was equally balanced with the drag force the flowrate required would be closer to 200 $\mu\text{l}/\text{min}$. However, for such a short DNA, the limited resolution of our low magnification lens might pose further challenges for detecting changes in projected length, which in this case may be on the order of 10s of nms. Furthermore, this scenario would result in substantial motion blur using the current data collection parameters and require a large motion blur correction factor. Therefore, work with shorter DNA would require a change in the setup allowing for higher camera bandwidth.

We would like to thank the reviewer for raising this important point. We have added a short summary of the issues that would arise with short DNAs at the end of the first results section.

REVIEWERS' COMMENTS:

Reviewer #1 (Remarks to the Author):

The Authors have addressed all my comments in a clear and thorough manner. The manuscript is suitable for publication.

Reviewer #2 (Remarks to the Author):

The authors have done a very good job in answering the various questions raised. I have only a very minor remark : to fit the WLC they use Eq. (2) page 21 line 529, this formula is an approximation which presents an error of 5 to 7 % at mid extension. The authors should at least warn the reader or use a more accurate expression.

The point-by-point response to the reviewer comments can be found below in blue after each point:

Reviewer #1 (Remarks to the Author):

The Authors have addressed all my comments in a clear and thorough manner. The manuscript is suitable for publication.

We are very happy the reviewer found our comments clear and thorough.

Reviewer #2 (Remarks to the Author):

The authors have done a very good job in answering the various questions raised. I have only a very minor remark : to fit the WLC they use Eq. (2) page 21 line 529, this formula is an approximation which presents an error of 5 to 7 % at mid extension. The authors should at least warn the reader or use a more accurate expression.

We are glad the reviewer pointed this out. We have added a warning for the reader and provided a reference to a more accurate expression as suggested at the end of the first paragraph of the methods section titled "Force determination."